# Does Selection for Longevity in *Acheta domesticus* Involve Sirtuin Activity Modulation and Differential Response to Activators (Resveratrol and Nanodiamonds)?

**DOI:** 10.3390/ijms25021329

**Published:** 2024-01-22

**Authors:** Patrycja Ziętara, Barbara Flasz, Maria Augustyniak

**Affiliations:** Faculty of Natural Sciences, Institute of Biology, Biotechnology and Environmental Protection, University of Silesia in Katowice, ul. Bankowa 9, 40-007 Katowice, Poland; patrycja.zietara@us.edu.pl (P.Z.);

**Keywords:** sirtuins, longevity, resveratrol, nanodiamonds, *Acheta domesticus*

## Abstract

Sirtuins, often called “longevity enzymes”, are pivotal in genome protection and DNA repair processes, offering insights into aging and longevity. This study delves into the potential impact of resveratrol (RV) and nanodiamonds (NDs) on sirtuin activity, focusing on two strains of house crickets (*Acheta domesticus*): the wild-type and long-lived strains. The general sirtuin activity was measured using colorimetric assays, while fluorescence assays assessed SIRT1 activity. Additionally, a DNA damage test and a Kaplan–Meier survival analysis were carried out. Experimental groups were fed diets containing either NDs or RV. Notably, the long-lived strain exhibited significantly higher sirtuin activity compared to the wild-type strain. Interestingly, this heightened sirtuin activity persisted even after exposure to RVs and NDs. These findings indicate that RV and NDs can potentially enhance sirtuin activity in house crickets, with a notable impact on the long-lived strain. This research sheds light on the intriguing potential of RV and NDs as sirtuin activators in house crickets. It might be a milestone for future investigations into sirtuin activity and its potential implications for longevity within the same species, laying the groundwork for broader applications in aging and lifespan extension research.

## 1. Introduction

Aging represents a multifaceted phenomenon discussed at various levels—populations, individuals, organs, cells, and molecules. This intricate process arises from external environmental factors and internal genetic influences [1]. Numerous theories highlight pivotal elements that expedite the aging process, including factors such as UV radiation, cellular oxidative stress, genetic mutations, underlying diseases, and even deficiencies in essential nutrients [2,3]. The subjects of anti-aging and longevity have perpetually intrigued the scientific community, spanning several decades of dedicated research. The extensive body of literature encounters numerous reports detailing novel particles, pharmaceutical compounds, bioactive substances, and even specific genes that offer valuable insights into the intricate mechanisms underlying lifespan extension. However, it is imperative to recognize that this field remains relatively underexplored and is the subject of ongoing research discourse. Consequently, it becomes imperative to search for general relationships and factors that hold the potential to unravel the complex puzzle of longevity, further advancing our comprehension of this intriguing phenomenon.

Considering recent studies, sirtuins (SIRT) emerge as significant players and can be regarded as a pivotal milestone in investigating aging biology. These signaling proteins constitute a noteworthy group closely linked with metabolic disorders. Thanks to their unique capacity for downregulating aging-associated genes, they are often perceived as “longevity enzymes” [4,5,6]. However, due to the complicated mechanism, the enzymatic functions of sirtuins have not yet been fully defined [7,8], but as far as we know, they are also involved in regulating the level of reactive oxygen species (ROS) by modulating the expression of antioxidant and detoxification enzymes [9,10]. Sirtuins are categorized from SIRT1 to SIRT7, and SIRT1 is currently (since 2023) recognized as the most extensively described, well-known, and thoroughly studied member of the sirtuin family. It plays a significant role in both developing and preventing neurological disorders. SIRT1 activates the AMPK pathway, the mTOR pathway, and the PPAR-γ pathway, which affects the secretion of adipokines and steroid hormones and modulates glucose and lipid metabolism [11,12,13,14]. Contemporary research has unveiled three primary sites of sirtuin localization: the cytoplasm, nucleus, and mitochondria. These distinct cellular compartments serve as key arenas where sirtuins exert diverse biological functions [4,15,16]. Sirtuins seem to be relatively widely described, but the complexity of the subject requires going beyond the framework and tackling many research issues. Information on sirtuin activity in cricket (a model species of incomplete metamorphosis) is so far unavailable, and current studies on invertebrates are limited to a few species like *Drosophila melanogaster* (predominantly), *Caenorhabditis elegans*, and *Ruditapes philippinarum* [4,17,18,19,20].

Numerous compounds have also been identified as potential activators of SIRT, but studying their mechanisms of action and discovering or modifying new activators presents challenges. Notably, these activators must be finely tailored to specific sirtuin–substrate pairings, introducing complexity to the research endeavors. Furthermore, selecting activator substances for sirtuins that adhere to stringent criteria, including safety, wide availability, non-toxicity, and high bioavailability, is imperative [4]. Research findings have proved that resveratrol (RV; 3, 5, 4′-trihydroxystilbene) acts as a sirtuin activator with specificity toward substrates. Moreover, RV can potentially influence an individual’s survival time [21,22,23]. In the appropriate concentration, it can extend the life span of *Saccharomyces cerevisiae* by up to 70% [24]. The retention time of RV in the cells depends on various factors, including dose, method of administration, metabolism, and enzyme activity in an organism. A study by Walle et al. [25] reported a half-life of RV in humans of approximately 9 h following a single oral dose of 25 mg RV, with an absorption capacity of 70%. After oral administration, Iannitti et al. [26] indicated that its plasma half-life in the human body ranges from 2 to 4 h. Regarding the specific formulation studied, Resv@MHD contains about 30% RV and 70% magnesium dihydroxide. This composition creates two types of microparticles: one of pure RV and the other of a complex with magnesium. This formulation demonstrates a different plasma profile compared to pure RV, including an earlier peak plasma concentration (25 min) and greater bioavailability. Additionally, a reported half-life of 80 min was observed with a dosage of 180 mg of pure RV [25,26].

Nanoparticles have become the subject of many studies, and scientists are looking for new ways to apply them, e.g., in medicine, biology, physics, or engineering [27]. Nanodiamonds (NDs) have garnered significant attention in diverse biomedical applications owing to their exceptional properties, including their diminutive size, expansive surface area, and compatibility with biological systems. These remarkable nanomaterials have undergone extensive investigation for purposes including drug delivery, tissue engineering, and bioimaging, among numerous other potential applications within biomedicine. Additionally, they have been shown to have low toxicity, making them a potential alternative to other nanomaterials that may have harmful effects. Overall, using NDs in biomedicine is a rapidly growing field with exciting potential for future advancements [28,29,30]. Some nanoparticles have demonstrated the capacity to augment the stability and bioavailability of diverse therapeutic agents, thereby facilitating precision-targeted and controlled drug delivery. Additionally, nanoparticle utilization holds the potential to diminish the requisite drug dosage, mitigating potential adverse effects and enhancing patient adherence to treatment regimens. While further investigations are imperative to comprehensively assess the safety and efficacy of nanoparticles in sirtuin modulation, they represent a promising avenue for developing innovative therapies targeting aging and age-related ailments and diseases. Nonetheless, it is noteworthy that nanoparticles may exhibit a limited cellular residence period, potentially limiting their effective interaction and complete realization of beneficial effects.

The main goal of this study was to characterize sirtuin proteins in the model insect species *Acheta domesticus* within two distinct strains, the wild-type (H) and the unique long-lived (D), and subsequently assess the impact of RV and NDs on their activity, considering selected physiological parameters, including those associated with lifespan extension. The following hypotheses were tested:

**H_1_.** 
*The SIRT selected for analysis can significantly differ between individuals from the wild strain (H) and the long-lived strain (D). It is more likely that individuals from the D strain exhibit higher activity in specific SIRT classes (all or part of them). However, the reverse relationship is also possible.*


**H_2_.** 
*There is a correlation between sirtuin activity, RV, or ND exposure and the lifespan of A. domesticus. Possible scenarios must involve not only the prolongation but also the shortening of the lifespan of individuals. RV may result in more noticeable effects for the D strain.*


**H_3_.** 
*There is a correlation between cell DNA damage and exposure to additional factors (NDs or RV). Such an effect may occur independently of changes in sirtuin activity. However, a positive correlation between disrupted sirtuin activity, impaired lifespan, and increased cellular damage/adverse effects is more likely.*


## 2. Results

### 2.1. Survival Analysis

Survival probabilities (Kaplan–Meier method) allowed for the illustration of the difference in survival between the studied groups (H and D strains). Individuals consuming food with RV exhibited the highest survival probability in both strains compared to the control and NDs groups. However, during a specific observation period, the survival rate in the NDs group in both strains was also higher than that in the control group (Figure 1a,b).

The Log-rank test was conducted to assess the significance of survival differences. Similar patterns were observed in both strains after the application of RV or NDs, leading to a significant increase in the survival probability of exposed crickets compared to the control group. Importantly, RV exhibited a more pronounced effect than NDs (Figure 1a’,b’). Additionally, when comparing groups from both strains, a significantly higher survival probability was observed in long-lived insects than in wild-type ones (Figure 1c’).

### 2.2. Sirtuin Activity in Acheta domesticus

The Brown–Forsythe test indicated homogeneity of variance in the groups (Figure 2a). Therefore, the results were analyzed using ANOVA. Results of the Factorial ANOVA (Figure 2b) indicate a significant main effect of all factors (‘Strain’, ‘Group’, and ‘Time’) as well as most of the interactions among them regarding the total sirtuin activity in the house cricket. The nonsignificant ‘Strain’ × ‘Group’ and ‘Group’ × ‘Time’ interactions indicate a similar pattern of SIRT activity in both strains after RV or ND treatment, which is similar over time.

On the sirtuins activity of *Acheta domesticus* in the H strain (Figure 2), it was observed that the control group maintained similar sirtuin activity values at all examined time points. RV stimulated SIRT activity in strain H, while NDs did not show this property. RV SIRT stimulation was most robust on day 20.

The results regarding sirtuin activity in the D strain (Figure 2) appear to correlate with the observed higher survival rates in this strain. Notably, the highest sirtuin activity values are achieved on the 15th day of observation in each tested group (control, RV, and NDs) compared to the 10th and 20th days. Interestingly, on day 20, stronger and more significant stimulation compared to the control was caused by NDs and not RV. Sirtuin activity on the 15th day remains higher than on the 10th and 20th days within treated groups (Figure 2c”), and the stimulation caused by RV was significantly higher than in the H strain at the same time point.

The results for the RV- and ND-treated groups were compared with the control group (depicted by the red line) in two strains (H and D). These measurements were taken over time. The presented data represent an averaged sample (Figure 3). Differences in the dynamics of the reaction were observed, dependent on the strain and the time and type of applied factor. These observations were confirmed by examining the interaction effects in a generalized linear model (GLM) (Figure 3a), indicating that time, strain, and group are substantial factors influencing the observed results.

Figure 3(b1–c3) shows that in strain H, the RV group on the 10th day exhibits a variable response of SIRT1, which is weaker than in the D strain. For the NDs group (10th day), the initial course of the reaction is similar in both strains; however, NDs exhibit slight stimulating effects from around 30 min after the start of the reaction. On the 15th day, we observe that the reaction to RV is similar in both strains (H and D), while NDs in strain H seem to show a greater effect than in strain D. On day twenty in the H strain, the response to RV and ND has the same dynamics, presenting an initial intense stimulation that declines over time. In the D group, the changes were not as spectacular as in the H strain, although RV stimulation is still evident.

When analyzing results concerning the interaction effect for sirtuin 1, a graphical interpretation of expected marginal means was carried out in the tested groups, focusing on the H strain (Figure 4(a1–a3)) and the D strain (Figure 4(b1–b3)).

A peak was observed in the H strain on the 10th day of the experiment (Figure 4(a1)), indicating a difference between the experimental groups. The control group exhibited a lower response than those treated with RV and NDs. On the 15th day (Figure 4(a2)), the marginal means for the RV and NDs groups began to converge, and by the 20th day, values for these groups increased, while the control group remained relatively stable.

A significant peak was also observed in the D strain on the 10th day (Figure 4(b1)), indicating a difference between the groups, like the H strain. The RV group achieved the highest marginal mean. On the 15th day (Figure 4(b2)), differences between groups (C, RV, and NDs) became more pronounced, especially for the RV group, which showed a significant increase in the marginal mean compared to the control group. On the 20th day, the marginal means for the control and RV groups indicated an increase compared to the levels observed on the 10th and 15th days.

To provide additional information regarding SIRT1 fluorescence, data were visualized using Relative Fluorescence Units (RFUs) (Figure 5). In the H strain, the control on the 10th day achieves a lower Relative Fluorescence Unit (RFU) than the control on the 10th day in the D strain, suggesting differences in the reaction dynamics between the strains. On the 10th day, in the RV and NDs groups in the H strain, initiation of the fast reaction was earlier than in control, with higher RFU values observed for the RV group. In the RV groups on the 15th and 20th days, the reaction onset appeared to be delayed compared to the 10th day, but NDs demonstrated a similar temporal pattern in these instances.

In contrast to these observations, in the D strain, the reaction started most rapidly in the RV-treated group, while in the ND-treated group, the dynamics of changes seemed to be similar to the control group.

### 2.3. DNA Damage

The results of the Brown–Forsythe test (Figure 6a) indicate no statistically significant differences in variances among groups for pATM, DSB, and pH2A.X on various days of the life cycle. The *p*-values for each parameter exceed the standard significance level of 0.05 on all three days (10th day, 15th day, and 20th day), suggesting homogeneity of variances across the studied groups.

The main effect of ‘Strain’ was not significant for the parameters pATM, DSB, and pH2A.X, indicating a similar range of values for these parameters in both strains of crickets. The ‘Group’ main effect was significant for DSB and pH2A.X, while the ‘Time’ main effect was significant only for DSB (Figure 6b). The interaction effects of ‘Strain’ × ‘Time’ differed significantly for a percentage of ATM-activated cells. For DSB, only the interaction of ‘Strain’ × ‘Time’ does not significantly impact. The interaction between ‘Strain’ × ‘Group’ and ‘Strain’ × ‘Time’ significantly affects pH2A.X.

The percentage of cells with activated ATM kinase (Figure 6) in the control groups did not differ between strains at any tested time point. RV caused a significant increase in pATM only in insects from the D strain on day 10 of adult life. However, when comparing the response to RV in both strains, a significantly higher level of pATM was observed in insects from the H strain compared to the D strain, but only on day 15 of adult life. Moreover, NDs increased the pATM level, but only in insects from the H strain on day 20 of adult life.

In the control group of strain H, the percentage of double-strand breaks (DSBs) (Figure 6) exhibited a decreasing trend over time at the examined points, mirroring similar observations in strain D insects. Additionally, on the 10th day of RV administration, a reduction in the value of DSB was noted. In the group consuming food with NDs, where initially the DSB levels were close to 40% for H and D strains on the 20th day, a decrease in these values occurred, reaching approximately 15% for both strains.

The percentage of H2A.X-activated cells (Figure 6) in the control group of strain H exhibited a twofold higher result compared to the control group of strain D on the 10th day. The administration of RV in the group consuming this compound significantly reduced the percentage of H2A.X-activated cells in strain H, similar to double-strand breaks (DSBs) (Strain H, 10th day). Moreover, the pH2A.X levels in the group with NDs in strain H showed only a slight decrease on the 20th day.

## 3. Discussion

### 3.1. Is the Life History of A. domesticus Significant for Sirtuin Activity?

Longevity, which is still a subject of open discussion, requires ongoing findings in interdisciplinary research as it represents a stage in the quest to identify common traits among different organisms and (typically) attempt to extrapolate these findings to humans [1,2,3]. Therefore, investigating the same species, even with distinctions such as different populations or, as in the case of crickets, various strains, emerges as a crucial aspect in unraveling the complexities of sirtuin activity. Commencing with survival analysis, a critical area of study in biology, we explore the factors influencing organism longevity. This investigation delves into survival data, unveiling the mechanisms that impact extended lifespan as our long-lived crickets indeed live longer and respond differently to SIRT stimulators.

The long-lived selected strain, as opposed to the H- strain, resulted from systematic efforts to obtain organisms with an extended lifespan, which has significant implications for understanding the impact of RV and NDs on survival. It is important to note that the decision to use this strain was non-random; rather, it was based on prior research and experiments carried out by our team [29,31]. The results of our study indicate that the selection for longevity is associated with a distinct pattern of SIRT activity in both strains. The main ‘Strain’ effect was significant for SIRT activity, which was higher in the long-lived strain (Figure 2b). Thus, the life history of this strain, including selection with the delay in reproduction, likely induced profound changes that support longevity at the level of regulatory proteins. Additional analysis of SIRT1 activity concerning its substrates, such as histones and transcription factors (e.g., p53, NF-κB, and FOXO), may help elucidate the observed changes [4,32,33]. Based on the information provided by Xuan et al. (2017), the fluorescence studies involving SIRT1 with the EGFP-K85AcK probe illuminate its role in enhancing fluorescence in *E. coli* ΔcobB, particularly in response to the acetylated lysine modification (HibK) at position K85 [34]. This would also indicate the involvement of SIRT1 in regulatory pathways in *Acheta domesticus*.

The observation that two distinct strains of *Acheta domesticus* (H and D) exhibit differences in survival underscores evolutionary adaptations in lifespan within a single species. Similar observations are also apparent in human populations’ lifespans. This indicates that specific genetic, environmental, or other factors in a population may impact organism longevity. In 2011, scientists identified long-lived populations, with the Nepalese being highlighted as one such group [35]. In recent years, it has been pointed out that the mortality rate for this country has decreased and the average lifespan in society has increased [36]. The research also points towards the polymorphism of SIRT3, which is more prevalent in long-lived populations [23].

Moreover, the differences in responses to RV or NDs among strains (H or D) underscore the complexity of genotype and environment interactions on the impact of bioactive substances or other factors in organisms, such as environment or even health status, as in humans [37]. This is particularly relevant given that May and Tomanek (2024) indicated in their research that environmental conditions in *Mytilus californianus* were one of the factors influencing total sirtuin activity [38]. The increase in sirtuin activity in response to NDs may suggest their potential beneficial impact on the regulatory ability of sirtuins, especially given that NDs are considered highly biocompatible [30]. However, observations suggest that RV may also induce different effects depending on the strain of *A. domesticus*.

The results regarding the total sirtuin activity prompted us to conduct a further analysis, including the study of SIRT1 fluorescence (Figure 3). The diverse reactions observed in different strains suggest different distinct dynamics in the stimulation of SIRT1. This variability may be attributed to factors such as age, specific stages of analysis, and the unique characteristics of each cricket strain, highlighting the intricate nature of RV or NDs’ impact on SIRT1 activity at the cellular level. In our previous studies, we observed variable responses in crickets at the level of oxidative stress, apoptosis, and DNA damage after exposure to graphene oxide [39]. Similarly, as in 2021, we indicated that prolonged exposure to NDs and differences between strains become noticeable after multigenerational exposure, whereas in the D strain, mobilization of the organism and activation of defense mechanisms were observed. This could be one of the factors influencing diverse cellular responses [29].

The decrease in double-strand breaks (DSBs) over time, especially in the D strain across all groups, suggests a potential reparative effect or reduced induction of DNA damage. Considering that DSBs are known to be cytotoxic, the observed reduction in their occurrence due to the tested substances implies potential therapeutic effects. However, the unexpected increase in DSBs after RV treatment in the H strain on day 15 indicates a dual role of RV in modulating DNA damage, possibly influenced by strain-dependent factors [40].

### 3.2. Is There a Correlation between the Age of Acheta domesticus and Sirtuin Activity?

The increase in SIRT enzyme activity does not always correlate with lifespan, although the Kaplan–Meier analysis demonstrates an association between increased activity of these enzymes and extended life. While heightened enzyme activity may contribute to favorable effects, excessive elevation could have adverse consequences [4,41]. In the context of our obtained results, the cellular parameters appear consistent because these suggest that the treatments have a time-dependent effect on the strains, with the NDs treatment showing a particularly dynamic response in the D strain. The differences between days could be due to the progression of the treatment effects or the adaptation of the strains to the treatments. An intriguing aspect is that on the 15th and 20th days in the D strain, RV appears to have a stimulating effect on SIRT1 activity, significantly expediting the initiation of the reaction. That is why the D strain appears to be more reactive to the treatments, compared to the H strain for strain D, and the 15th day appears to be the optimal window of sensitivity to stimulators in the form of RV and NDs.

However, the impact of RV and NDs on SIRT1 activity may not be only contingent upon the reaction initiation time (Figure 5) but also on specific regulatory mechanisms, varying between the analyzed strains (H or D) and, as previously mentioned, measurement days. The actions of these substances can exhibit diversity and dynamics, leading to disparate effects at different stages of the experiment. As the organism ages, metabolic changes may impact sirtuin activity [23]. Given their involvement in DNA damage repair, the quantity and types of damage in genetic material may increase with aging, necessitating greater sirtuin activity in repair processes. These assumptions appear to be supported by Peng et al. (2015), which indicated a direct association between SIRT1 and DNA damage repair [42].

In our research, the observed fluctuations in pATM activity in the H strain, especially the significant increase on day 15 after RV treatment, suggest a dynamic response to the substance over time. This contrasts with the D strain, where the highest pATM activity was recorded in the RV group on day 10, indicating potential strain-dependent differences in the temporal dynamics of the response to DNA damage. This phenomenon indicates that the ATM kinase plays a crucial role in the early stages of the cellular response to double-strand breaks (DSBs) in DNA. This kinase performs a key function by phosphorylating the H2AX protein, a significant step in the cellular response to DNA damage. ATM is a major regulator of DNA repair mechanisms, particularly in situations involving severe DNA damage, such as double-strand breaks. This phenomenon is critical to understanding how cells respond to such damage and initiate repair processes [40].

The D strain increases H2A.X activity in the control group, indicating potential accumulation of DNA damage with the insect’s age. Notably, the ND group in the D strain maintains relatively stable H2A.X activity, aligning with the RV group on days 10 and 15. This consistency maintenance raises questions about the potential protective role of NDs against age-related DNA damage in the D strain. However, in our recent studies, we suggested that NDs could stabilize DNA by interacting with chromatin, forming a compact structure, and potentially influencing chromatin-stabilizing proteins. This may incidentally limit the amount of DNA damage [43].

In addition, it covered a selected period of life (10, 15, and 20 days of imago), which may not be crucial for sirtuins in *A. domesticus*. For this reason, it is also necessary to check the activity in other stages.

### 3.3. Does the Resveratrol and Nanodiamonds Influence Sirtuin Activity?

Discussions regarding the safety of using NDs in various scientific fields (e.g., biology, medicine, genetics, and chemistry) are ongoing [44,45]. The direct correlation has no clear-cut pieces of evidence or studies suggesting that NDs possess such an ability. Nevertheless, research into the use of NDs as drug carriers or other substances for therapeutic purposes is underway. Such applications may aid in treating various ailments and improving health, potentially influencing the lifespan of individuals affected by specific diseases [46,47,48]. RV is known for its potential health benefits that may influence longevity [4]. It can be perceived as a general activator of sirtuins, influencing the regulation of metabolic processes and the organism’s aging [49,50]. Current studies have confirmed this. However, crickets from the NDs group also exhibited an increased percentage of survival probabilities, which suggests that responses to NDs are not random.

In our previous research, we discussed the complex response of *A. domesticus*, whereby graphene oxide exposure involves intricate adaptive mechanisms. The study underscores the importance of considering multigenerational effects and the potential trade-offs between short-term adaptations and long-term impacts on survivability and genomic stability [51]. Notably, Mytych et al. (2016) reported that low concentrations of nanoparticles, including NDs, may lead to increased expression of SIRT1 while simultaneously activating cellular stress response mechanisms [45]. That is why the presence in the diet or activation during the strain selection process may lead to increased sirtuin activity because the observed differences in the dynamics of sirtuin 1 activity were dependent on the strain, time, and the type of applied factor. These findings were confirmed by examining interaction effects in GLMs, highlighting the substantial influence of time, strain, and group on the observed results. Moreover, among the studies carried out by other researchers, varying effects on lifespan extension through RV have been demonstrated. For example, in studies on *Nothobranchius guentheri*, RV not only delayed aging but also provided protection against neurodegeneration [52]. Conversely, Song et al. (2021), researching *Bombyx mori*, demonstrated an activation of the SIRT7-FoxO-GST signaling pathway and an extension of lifespan following treatment with RV. Their studies also conducted a lifespan assay, indicating a connection between RV and longevity [53]. Current research provides a wider insight into the interaction mechanism, not only of RV but also NDs, with sirtuins in *Acheta domesticus*.

Nevertheless, on the 10th day, stimulation of SIRT1 activity was observed under the influence of NDs and RV in both strains (H and D). This suggests that the impact of NDs and RV on SIRT1 regulation is more complex and may vary at different stages of the organism’s response to their presence. Such complex interactions between RV and NDs may be related to interactions with other regulatory factors in organisms, such as longevity-associated genes, metabolic pathways, or stress factors [29,48]. These significant differences in responses to RV and NDs between the H and D strains were also observed in the analysis of their survival. As mentioned earlier, these distinct responses are suspected to arise from genetic, molecular, and regulatory aspects of the cells. However, it is important to note that nuclear extracts can identify the general presence of SIRT1, SIRT6, and SIRT7 in the analyzed tissues.

The activation of histones (ATM and H2A.X) (Figure 6) in tested groups may indicate the initiation of repair mechanisms, as evidenced by the decreasing trend in double-strand breaks relative to the measurement points (10, 15, and 20 days) and the aging of individuals. Similar conclusions were drawn in our previous studies on NDs [29]. Furthermore, NDs at the applied dose appear not to induce a toxic effect, which aligns with recent studies emphasizing their relatively low toxicity and facile biodegradation, as highlighted by Bilal et al. (2021) [54]. A similar trend is observed for the group treated with RV. This also allows us to infer that RV may exhibit a potentially therapeutic effect, which could be linked to the activation of sirtuin or/and a specific sirtuin.

In summary, these results suggest that RV and NDs exert a diverse and dynamic impact on the activity of SIRT1 in the examined strains. This, in turn, may influence various regulatory processes related to longevity, providing an important avenue for further research and a better understanding of these interactions. Research into possible interaction between SIRTs, NDs, and RV might unveil new insights into the metabolic processes governing organismal longevity. The findings from this study can be pivotal in guiding future experiments, not only on other animal models but also potentially in clinical settings. With appropriate ethical considerations and optimizations, this experimental model could be extended to vertebrates, paving the way for clinical trials exploring RV as an aging process inhibitor and marking a step toward translating laboratory findings into clinical applications. Further research is needed to fully understand these interactions and their potential implications for the health and longevity of organisms consuming RV and NDs.

## 4. Materials and Methods

### 4.1. Nanodiamonds

We employed NDs commercially known as Single-Digit NDs in the study, grade “SDND” 5 wt.% aq. suspension, obtained from PlasmaChem GmnH, Berlin, Germany. As reported by the manufacturer, these nanodiamonds were produced through chemical disintegration and are devoid of additives and milling impurities. The diamond crystallite size ranges from 3.5 to 5.2 nm, and the particle size, as determined by Dynamic Light Scattering (DLS), falls within the range of 5 to 15 nm. These NDs underwent characterization as described in the article [55]. In the research carried out by the team in 2022 [29], a comprehensive analysis revealed the gentle aggregation of NDs within the suspension. However, a negative and relatively low zeta potential (−15.6 mV, 25 °C) proved good ND stability in the suspension. The NDs were also visualized using AFM (Agilent 5500 Atomic Force Microscopy, Agilent Technologies, CA, USA), confirming that the size of the NDs is consistent with the scale declared by the manufacturer, with results ranging from 2.8 to 5.5 nm (Figure 7). A detailed characterization can be found in the studies by Augustyniak et al. (2023), where the same material was employed for the research [43].

### 4.2. Resveratrol

The experiments employed RV (CAS 501-36-0; molecular weight: 228.24 g/mol) sourced from Pol-Aura^®^. As indicated by the manufacturer, this compound has a natural origin. The Certificate of Analysis (COA) confirms the compound’s high purity, as evidenced by its ≥98% content according to high-performance liquid chromatography (HPLC) analysis.

### 4.3. Experimental Design

#### 4.3.1. Characteristics of *Acheta domesticus*

The insect species used in our research was *Acheta domesticus*, a member of the family *Gryllidae* within the order Orthoptera. This species, commonly known as the house cricket, is widely distributed and frequently encountered in human habitats worldwide. It undergoes several developmental stages in its life cycle, starting as an egg, progressing through multiple nymphal stages resembling miniature adult forms without fully developed reproductive organs, and culminating in the adult stage (IMAGO) with fully developed wings and reproductive capabilities. The lifespan is around 3–4 months. Due to its distinctive attributes and widespread presence, house cricket has garnered significant interest from researchers across diverse fields. It is widely used in toxicological, physiological, and genetic research and is considered a model organism because it is a relatively easy organism to cultivate [56]. We have been cultivating this species for nearly 30 years [29], providing us with extensive knowledge and experience. In the present study, we utilized two strains of house crickets: the wild-type (designated as H) and the long-lived (designated as D). The long-lived strain was obtained through selection lasting many years, which involved postponing reproduction.

The long-lived strain is characterized not only by extended longevity but also by a delayed onset of reproduction compared to the wild-type strain upon reaching the imago stage. Due to its high reproductive capacity, the house cricket allows enough individuals for research purposes in a short time [51,57].

#### 4.3.2. Experimental Model

All experiments were carried out within tightly controlled environmental conditions, including a temperature of 28.8 ± 1 °C, a photoperiod of 12 h of light and 12 h of darkness (L:D 12:12), and a humidity level of 43.80 ± 6.72%. Insects were fed standardly by providing finely ground “Rabbit Fattening Feed KDT, UNIPASZ; granulate form” [ingredients: wheat bran, sunflower seed cake, sunflower seed meal, wheat, beet molasses, triticale, calcium carbonate, calcium-magnesium carbonate, sodium chloride, family oils and fats, sodium salts of organic acids, herbal extracts]. This feed was subsequently blended with either RV or NDs in appropriate proportions to achieve the desired concentrations of RV or NDs in food. After blending, the mixture was dried, sterilized, and stored in containers until use.

Subsequently, freshly hatched insects from both strains H and D were categorized into three distinct groups: the control group (receiving food without additives), the RV group (RV in concentration 23 mg kg^−1^ in food), and the NDs group (NDs in concentration 0.2 mg kg^−1^). Each group was consistently fed with their respective diets from hatching throughout the experiment, which lasted for 56 days. After the insects reached the imago stage, five individuals were randomly selected from each experimental group at three time points: 10, 15, and 20 days (Figure 8). To prepare insect tissues, individuals were first anesthetized on ice before measurements were taken. As we have previously detailed in our prior articles [29,51,57], cell suspensions derived from the house cricket intestine were prepared in 0.1 M PBS (pH 7.4) and homogenized using the Minilys homogenizer (Bertin Technologies, Montigny-le-Bretonneux, France).

### 4.4. Biochemical Analysis

#### 4.4.1. Extraction of Cell Nuclei and SIRT Activity

Before commencing the procedure for detecting sirtuin activity in *A. domesticus* tissue, nuclear extracts were prepared using the Nuclear Extraction Kit (Abcam, Cambridge, UK, ab113474), recommended by the manufacturer of the Universal SIRT Activity Assay Kit (Colorimetric) (Abcam, Cambridge, UK, ab156915) test. The Nuclear Extraction Kit employs a 1000× Protease Inhibitor Cocktail, 10X Pre-Extraction Buffer, DTT Solution (1000×), and ENE2 (Extraction Buffer) to rapidly isolate abundant nuclear extracts within a relatively short assay time of approximately 1 h. This was used for *A. domesticus* intestine tissues, and then the extracted nuclear components were promptly employed for experimentation.

Activity Assay Kit (colorimetric) (Abcam, Cambridge, UK, ab156915) measured the overall SIRT enzyme activity in the previously prepared nuclear extracts. The kit incorporates a deacetylated histone standard, simplifying the quantitative evaluation of the activity of SIRT enzymes. Samples were measured using a UV–vis spectrometer (TECAN Infinite M200, Tecan Austria GmbH, Grödig, Austria).

The intensity of the SIRT1 reaction was assessed in nuclear extracts obtained using the SIRT1 Activity Assay Kit (fluorometric) (Abcam, ab156065). Measurements were performed using a Fluorescence Spectrometer Plate Reader (HITACHI F-7000, Hitachi, Ltd., Tokyo, Japan) over a 60 min duration. This assay facilitated the detection of SIRT1 activity.

#### 4.4.2. DNA Damage

The research kit employed, Muse^®^ Multi-Color DNA Damage Kit (Luminex Corporation, TX, USA, Part Number: MCH200107), enabled the detection of ATM and H2A.X protein activation, which is integral in the ATM-dependent signaling pathway associated with DNA damage response. ATM kinase activation and H2A.X phosphorylation are key indicators of cellular responses to DNA damage, especially in the context of double-strand break repair. The test provides percentages encompassing cells without DNA damage (negative cells), ATM-activated cells, H2A.X-activated cells, and cells with double-strand DNA damage (dual activation of both ATM and H2A.X).

The method, relying on the simultaneous use of two antibodies conjugated with fluorochromes, enabled the analysis of the phosphorylation status of ATM (Ser1981) and Histone H2A.X. This approach consequently allowed for the examination of signaling pathways associated with DNA damage repair. The following steps of the testing procedure were conducted in accordance with the manufacturer’s recommendations. Cells were prepared, fixed, permeabilized, and then incubated with an antibody cocktail specific for ATM and H2A.X. Following incubation, the cells were washed, resuspended in assay buffer, and analyzed using the Muse^®^ Cell Analyzer to quantify DNA damage indicators.

### 4.5. Statistical Procedures

The non-parametric statistical test, Kaplan–Meier survival analysis, was employed to assess survival rates in the experimental groups. Survival curves depicting the probability of survival across various time intervals were graphically presented. Subsequently, the Log-Rank Test (Chi-squared, *p* < 0.05), particularly valuable for evaluating differences in survival, was carried out.

SIRT activity and DNA damage across three time points (10th, 15th, and 20th days of the imago stage of *A. domesticus*) were measured five 5 replicates per group (control, RV-, and ND-treated). The Levene and Brown–Forsythe tests were used to check differences in the homogeneity of variance between the studied groups. To identify differences between experimental groups and strains, an ANOVA (LSD test, *p* < 0.05) was employed using STATISTICA^®^13 software (TIBCO Software Inc., Palo Alto, CA, USA). Univariate factorial ANOVA and interaction effects were also performed for total SIRT activity and DNA damage using sigma-constrained parameterization. GLM analysis with consideration of the interaction effect for SIRT1 was performed.

## 5. Conclusions

Our study not only represents a significant step in understanding the impact of RV and NDs on organisms, including *A. domesticus,* but also focuses on the sirtuin activity within the same species across two different strains. One of these strains was deliberately selected for longevity, while the other remained wild-type.

The results of our study indicate a diverse impact of RV and NDs on SIRT1 activity, depending on the strain and the life stage of the organisms. The reduction in SIRT1 activity in the D strain may be associated with regulatory mechanisms linked to longevity. Simultaneously, the observation of SIRT1 activity- and stimulation-specific conditions (depending on age, strain, or time) suggests that the influence of these substances may be more complex and dynamic.

It is worth emphasizing that there are potential therapeutic implications arising from these observations. The action of RV may prove beneficial to the health of organisms. However, due to the complexity of the impact of these substances on organisms, further studies at different life stages are necessary for a better understanding of these relationships and their potential implications for extending lifespans. Our research opens new perspectives in the study of the effects of RV and NDs on aging processes and organism health, constituting an important field of research in the biology of aging.

## Figures and Tables

**Figure 1 ijms-25-01329-f001:**
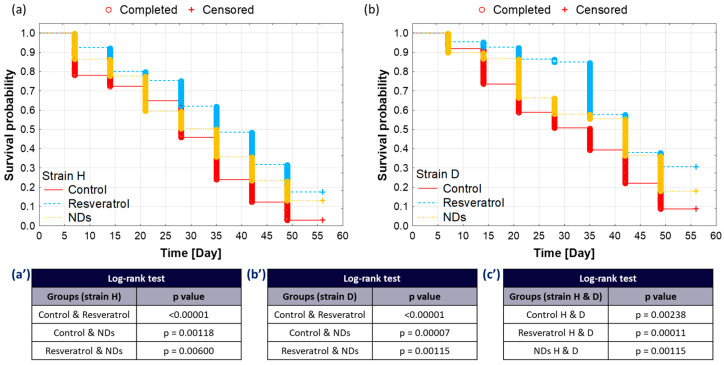
Kaplan–Meier survival analysis: (**a**) for the wild-type strain (H) over 56 days; (**b**) for the long-lived strain (D) over 56 days. (**a’**,**b’**) Log-rank test results comparing survival distributions within the respective strain. Experimental groups: control-free food; resveratrol group (23 mg/kg in food); and nanodiamonds group (0.2 mg/kg in food). (**c’**) Comparison of the same study groups between strains H and D (STATISTICA^®^13, TIBCO Software Inc., Palo Alto, CA, USA).

**Figure 2 ijms-25-01329-f002:**
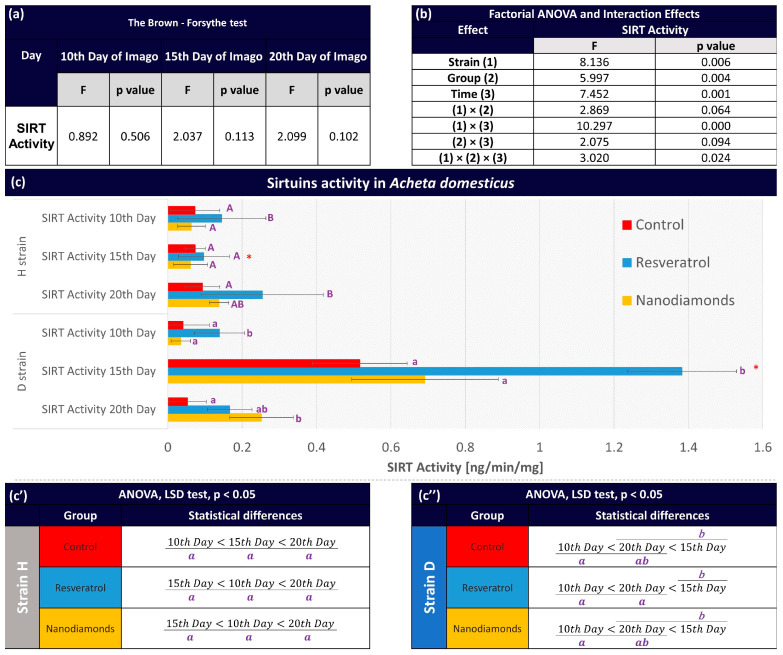
(**a**) Results of the Brown–Forsythe test for SIRT activity within days. (**b**) Results of a factorial ANOVA assessing the effects of strain, group, time, and their interactions on SIRT in *Acheta domesticus*. (**c**) The activity of sirtuins was expressed as mean ± SD (ANOVA, *p* < 0.05). Capital letters denote homogenous groups within the H strain and the same measurement day. Lower-case letters denote homogenous groups within the D strain and the same measurement day; “*” Differences between strains (H and D). (**c’**,**c”**) Within-group comparisons of SIRT activity across different measurement days (10th, 15th, and 20th of the imago stage) for H (**c’**) and D (**c’’**) *A. domesticus*. Statistical differences between the days within each group are denoted by distinct lower-case letters. Groups that share the same letter are not significantly different (ANOVA, LSD test, *p* < 0.05).

**Figure 3 ijms-25-01329-f003:**
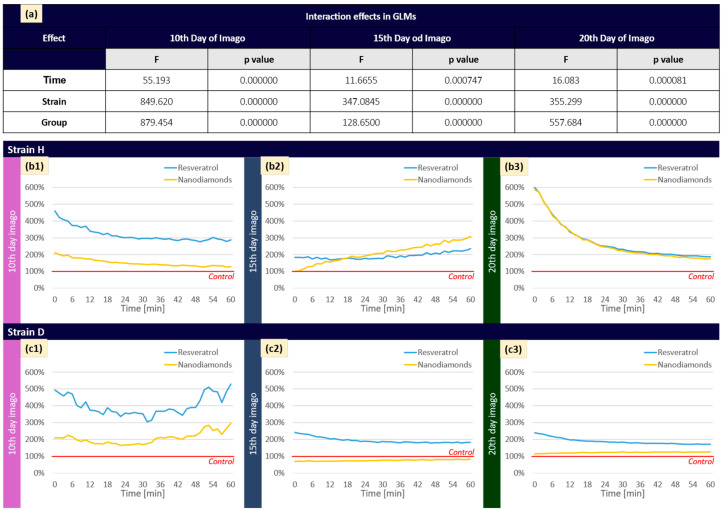
(**a**) Interaction effects in GLMs. (**b1**–**b3**) Mean percent of SIRT 1 fluorescence in *Acheta domesticus* over 60 min within time points of measurements (10th, 15th, and 20th day of imago stage, respectively) in H strain. (**c1**–**c3**) Mean percent of SIRT 1 fluorescence in *A. domesticus* over 60 min within time points of measurements (10th, 15th, and 20th day of imago stage, respectively) in D strain. (**b**,**c**) Control was presented as 100% (red line), and the results for the RV or ND-treated groups were recalculated accordingly.

**Figure 4 ijms-25-01329-f004:**
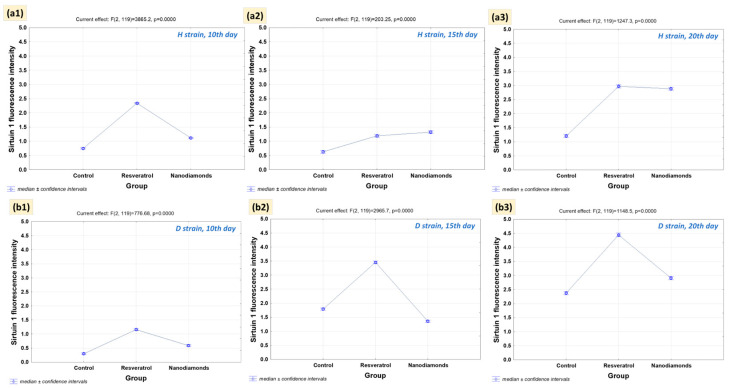
Graphical interpretation of the interaction effect for sirtuin 1. Expected marginal means in tested groups at three time points (10th, 15th, and 20th days of the imago stage). (**a1**–**a3**) H strain and (**b1**–**b3**) D strain. Vertical bars represent 0.95 confidence intervals (GLM, STATISTICA^®^13, TIBCO Software Inc., Palo Alto, CA, USA).

**Figure 5 ijms-25-01329-f005:**
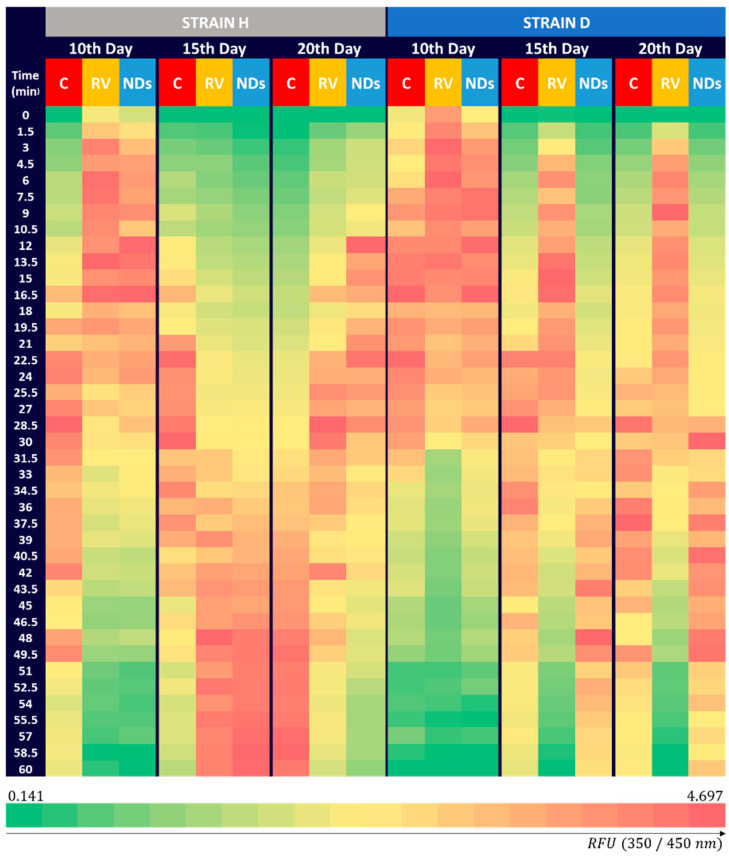
Mean RFU (Ex = 350 nm, Em = 450 nm) SIRT 1 fluorescence in *Acheta domesticus* over 60 min in two strains (H and D) and groups tested (control—C; resveratrol—RV; nanodiamond—NDs) at three time points (10th, 15th, and 20th day of imago stage).

**Figure 6 ijms-25-01329-f006:**
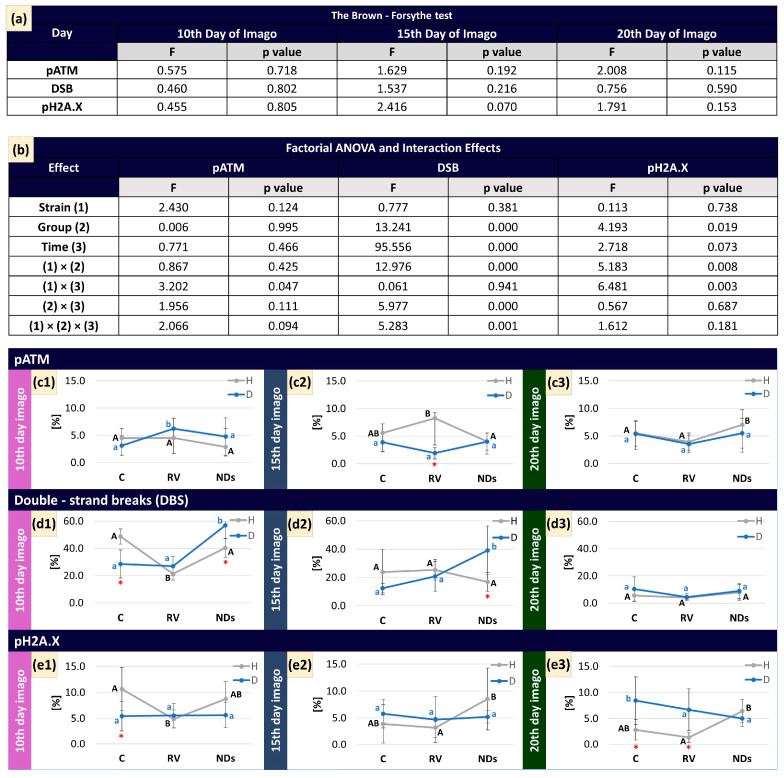
(**a**) Results of the Brown–Forsythe test. (**b**) Results of a factorial ANOVA assessing the effects of strain, group, and time, and their interactions, on SIRT in *Acheta domesticus*. The activity of sirtuins is expressed as mean ± SD (ANOVA, *p* < 0.05). (**c1**–**c3**) ATM activated cells (pATM) measured on 10th, 15th, and 20th time points, respectively; (**d1**–**d3**) double-strand breaks (DSBs) measured on 10th, 15th, and 20th time points, respectively; and (**e1**–**e3**) histone H2A.X-activated cells (pH2A.X) measured by Muse^®^ Multi-Color DNA Damage on 10th, 15th, and 20th time points, respectively. (**c**–**e**) Mean ± SD are shown in the charts. The same capital letters denote homogenous groups within the H strain; the same lower-case letters denote homogenous groups within the D strain; “*” means differences between groups within different strains (ANOVA, LSD test, *p* < 0.05).

**Figure 7 ijms-25-01329-f007:**
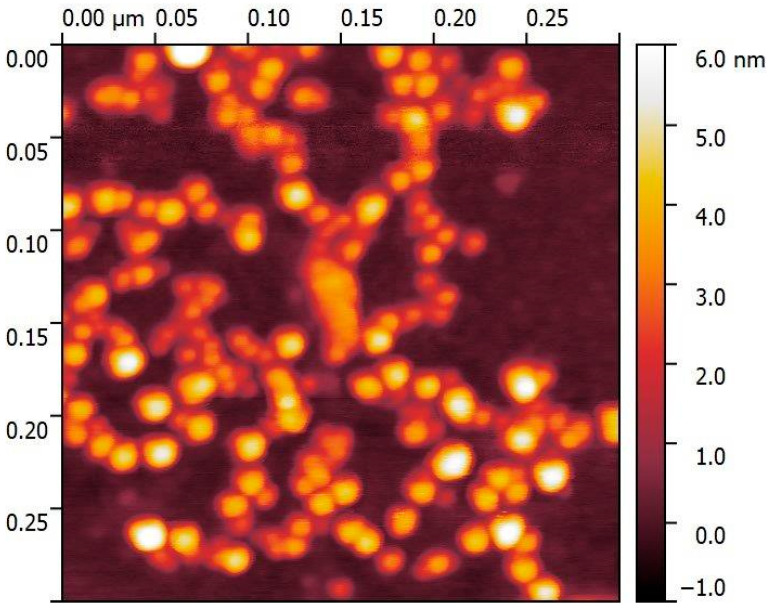
AFM of nanodiamonds analysis (Agilent 5500 Atomic Force Microscopy).

**Figure 8 ijms-25-01329-f008:**
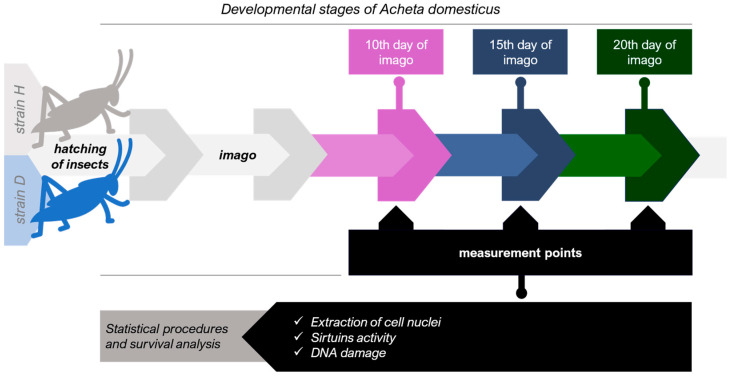
Experimental Design.

## Data Availability

Data are contained within the article.

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
