# Peer review of "Does Selection for Longevity in Acheta domesticus Involve Sirtuin Activity Modulation and Differential Response to Activators (Resveratrol and Nanodiamonds)?"

_ijms, 2024, doi:10.3390/ijms25021329_

Round 1
Reviewer 1 Report
Comments and Suggestions for Authors
I have been carefully evaluating the manuscript entitled “Does selection for longevity in Acheta domesticus involves Sirtuin activity modulation and differential response to activators (Resveratrol and nanodiamonds)?“ by Zietara and colleagues. The study reports the impact of resveratrol and nanodiamonds on Sirtuins activity using the wild-type and long-lived strains of house crickets.
Although the topic is very interesting, the manuscript does not reach enough relevance and priority to warrant publication in IJMS. This consideration is based on various aspects including the lack of functional assays and the difficult to understand some tables and graphs. Globally, the results are uncertain and inconclusive.
Here the main criticisms:
1) Sirtuins activity and DNA damage were only analysed by weak biochemical assays. They should be further supported by alternative experiments (such as western blot on gamma-H2AX, pATM and acetylation of histone and transcription factors).
2) For the more transparent and comprehensible communication of scientific results, it should be appropriate for the authors to report real p-values of their analysis and to convert several tables (2b, 2C’, 2C” and 6b) in graph bars.
3) The number of house crickets used in each experiments should be reported in the table legends
Comments on the Quality of English LanguageModerate editing of English language is required
Author Response
As in the attached file.

Reviewer 2 Report
Comments and Suggestions for Authors
1. Please give some information (lifespan, stages and related duration, and advantage of using them) about the animal model (house cricket) in the method or introduction section.
2. Lines 73-76, the case of Walle et al’s study should specify the experimental subject; the case of Iannotti et al’ study should specify the intake dosage and formulation.
3. Lines 132-133, the concentration of resveratrol and nanodiamonds are expressed in different unit, what is the reason to not unify them?
4. Line 166, what is “Survival For sirtuin 1 the results for the RV- and NDs-treated groups” ?
5. Line 171, full name of ‘GLMs’? general linear model?
6. It seems lots of the abbreviated terms are not explained inside the manuscript. Any plan to make a list of abbreviations with their full names?
7. Fig. 3, what is b1, b2, b3, c1, c2, c3? Different days? Please add the information into the caption of this figure and the rest of figures with similar issues in the manuscript.
8. Please don’t use the full forms of NDs and RV (nanodiamonds and resveratrol) once they have been abbreviated in the manuscript. Please check these issues throughout the manuscript.
9. Line 493, it says the insects were selected at three time points: 5, 10, 15 days. But samples from 10, 15, 20 days were used in the rest manuscript. Which one is correct??
10. Please unify the format of reference, especially about the format of the journals’ titles.
11. Please give some more information about the reason using the kits evaluating ATM and H2A.X protein activation to determine DNA damage. The test procedure recommended by the supplier should be briefly described in the method section.
12. Please specify the starting/ending time points of intervenation using RV/NDs in the method section.
13. please give some more discussion about the reference significance of the current dynamic outcomes of RV/NDs-administrated house cricket model to the future experiments on other animal and even clinical studies.
Comments on the Quality of English Language
Most of the manuscript are in a good shape
Author Response
As in the attached file.

Round 2
Reviewer 1 Report
Comments and Suggestions for Authors
no further comments